# SFP: Spurious Feature-Targeted Pruning for Out-of-Distribution Generalization

Submission Id: 2007*

## ABSTRACT

Recent studies reveal that even highly biased dense networks can contain an invariant substructure with superior out-of-distribution (OOD) generalization. While existing works commonly seek these substructures using global sparsity constraints, the uniform imposition of sparse penalties across samples with diverse levels of spurious contents renders such methods suboptimal. The precise adaptation of model sparsity, specifically tailored for spurious features, remains a significant challenge. Motivated by the insight that in-distribution (ID) data containing spurious features may exhibit lower experiential risk, we propose a novel **S**purious **F**eature-targeted **P**runing framework, dubbed **SFP**, to induce the authentic invariant substructures without referring to the above concerns. Specifically, SFP distinguishes spurious features within ID instances during training by a theoretically validated threshold. It then penalizes the corresponding feature projections onto the model space, steering the optimization towards subspaces spanned by those invariant factors. Moreover, we also conduct detailed theoretical analysis to provide a rationality guarantee and a proof framework for OOD structures based on model sparsity. Experiments on various OOD datasets show that SFP can significantly outperform both structure-based and non-structure-based OOD generalization state-of-the-art (SOTA) methods by large margins.

## KEYWORDS

Out-of-distribution Generalization, Model Pruning, Deep Neural Network, Module Detection

**ACM Reference Format:**

Anonymous Author(s). 2024. SFP: Spurious Feature-Targeted Pruning for Out-of-Distribution Generalization. In *Proceedings of (ACM Multimedia 2024)*. ACM, New York, NY, USA, 18 pages. https://doi.org/XXXXXXX.XXXXXXX

## 1 INTRODUCTION

Deep neural networks trained with empirical risk minimization (ERM) [33] learn correlated features thoroughly to achieve superior accuracies. However, when confronted with fickle real-world data distributions, even a slight shift renders most applications vulnerable due to the idealistic assumption that the data are identically and independently distributed (IID). There are several reasons for this failure: firstly, if data are generated from a fully observed

causal Bayesian network (CBN), ERM would learn all features in the Markov blanket, even those not causally related [4, 5, 41]. Secondly, substantial works have demonstrated that ERM's prediction tends to exploit spurious correlations or shortcuts that are prone to change in real-world distributions [8, 9, 26]. Hence, essentially understanding and restraining the learning of spurious correlations is crucial.

Out-of-distribution generalization, which focuses on learning causally correlated features that remain invariant across different domains, has received significant attention. Most recently, a line of work set out to improve OOD generalization from the perspective of model structure. Can models with particular structures avoid neural networks being biased towards spurious correlation in out-of-distribution (OOD) generalization [41]? Most works provide a positive answer. For example, Sagawa *et al.* [32] provides sufficient and intuitive motivation for this branch, claiming that over-parameterized models could degrade OOD performance through data memorization and overfitting. Zhang *et al.* [41] have a similar conclusion. They articulate and demonstrate the functional lottery ticket hypothesis: full network contains a subnetwork that can achieve better OOD performance. Compared to typical causal representation learning, structural approaches have extra benefits of universality and efficiency. Most works can be embedded in non-structural SOTAs to generate slimmed networks with better OOD performance.

Despite substantial advancements, existing structural methods are predominantly designed empirically and lack theoretical interpretability. It has been observed that these approaches typically use established techniques in a rudimentary manner without specific refinements to unearth OOD lottery tickets, including network architecture search, module detection, and model pruning. This may fail to pinpoint the optimal OOD structure due to the imposition of global sparsity constraints. More precisely, many studies enforce equal parameter penalties for learning across diverse features. As an illustration, Sagawa *et al.* [41] explicitly state that the sparsity of structures does not exactly correspond to the sparsity of spurious features in their method. Except for improper optimization objectives, the majority of these approaches rely on the guidance of fully exposed OOD datasets, which is essentially infeasible in real-world applications.

To address these issues, we propose a novel **S**purious **F**eature-targeted network **P**runing method, dubbed **SFP**, to explore the optimal OOD substructures. The key idea is to selectively impose optimization constraints to prevent the leakage of spurious features into the learned patterns. Specifically, SFP employs meticulously derived thresholds from training dynamics, enabling it to discern biased samples entangled with spurious correlations during the training phase. Following this discernment, SFP seamlessly incorporates the feature projection onto the model space as a regularization term, effectively reining in the model's alignment with

specific feature directions. Extensive experiments conducted on various datasets have demonstrated that the proposed SFP achieves superior performance than most of the state-of-the-art methods.

In summary, our contributions can be outlined as follows:

- We propose a theoretical framework that substantiates the rationale and effectiveness of improving OOD generalization through feature-specific model sparsity. This contribution serves to address the deficiency of theoretical guidance present in prior research within this domain.
- We propose a novel spurious feature-targeted model pruning to explore OOD substructures, totally without prior causal assumptions or full exposure of out-domain data.
- To our knowledge, we are the first to theoretically unveil the adjustable correspondence between data features and model substructures within OOD settings, as well as leverage it to enhance the generalization performance.

## 2 RELATED WORK

**Out-of-Distribution Generalization.** Existing research on OOD generalization can be roughly divided into two categories, including non-structure-based methods and structure-based methods. Specifically, the non-structure-based methods focus on the feature level and usually limit models over learning on spurious features by designing heuristic learning paradigms or separating different features in high dimensions. For example, Arjovsky *et al.* [2] aims to extract nonlinear invariant predictive features across multiple environments. IIB [18] performs invariant feature prediction by limiting the mutual information between the learned representation and the ground truth. While effective, unstructured methods yield only partial benefits from representation learning, resulting in an over-parametric final model that may compromise generalization performance. Differently, the structure-based methods investigate the impact of different modules on OOD generalization. Early work can be traced back to [29], which affirms that models with specific structures under linear conditions can avoid false correlations in OOD generalization. Most recently, Zhang *et al.* [41] proposes the functional lottery hypothesis, which further confirms the improvement of model structure on OOD generalization performance under OOD setting and nonlinear condition. Moreover, this positive impact can be superimposed on most previous non-structure-based methods. *However, these methods directly utilize model compression algorithms while ignoring the relationship between data features and model structures, potentially leading to suboptimal results.*

**Model Pruning.** A series of network pruning methods have been proposed to eliminate unnecessary weights from over-parameterized networks. Early research [17] usually tries to remove weight parameters based on the Hessian matrix of the objective function. Similarly, Han *et al.* [11] proposes to remove the weights or nodes with small-norm from DNNs. However, these kinds of unstructured pruning (i.e., discrete weights or nodes) can hardly reduce reasoning time without specialized hardware [38]. Therefore, structured pruning [20, 38], i.e., channels/filters, is more applicable and becomes mainstream. For example, He *et al.* [12] resets less important filters at every epoch while updating all other filters. Zhao *et al.* [43] uses stochastic variational inference to remove the channels with smaller mean/variance. *Despite all that, previous methods essentially follow the traditional empirical risk-guided model pruning paradigm; thus, the obtained feature-untargeted sparse model is suboptimal for OOD generalization.*

## 3 PROPOSED METHOD

We start by formalizing the model structure-based OOD problem in a complete *inner product space* and then provide a theoretical analysis to investigate the impact of ID data and out-domain data on model performance. Based on this framework, we elaborate on the optimization objective of SFP and theoretically demonstrate its effectiveness.

### 3.1 Notations and Preliminaries

*3.1.1 Linear Parameterized Notations.* Let $X_{id} \in \mathbb{R}^{p \times d}$ and $X_{ood} \in \mathbb{R}^{q \times d}$ be the in-domain and out-domain datasets, respectively, where $p$ and $q$ denote the numbers of data instances, and $d$ is the feature dimension. Consequently, the entire training dataset can be represented as $X = X_{id} \cup X_{ood}$, where $X \in \mathbb{R}^{n \times d}$ with $n = p + q$. The corresponding ground truth of the feature projection is represented by $Y$. Additionally, let $p_i$ and $p_o$ signify the proportions of instances with and without spurious features in the training set, respectively, such that $p_i + p_o = 1$. To rigorously elucidate our analysis and proofs, we align with the theoretical framework established by previous works [6, 16, 37]. Specifically, they consider a linear format for the feature extractor and define logits as the projection length of input onto a specific subspace. Based on the "implicit regularization effect of initialization [28]" and the "deep multi-layer homogeneity [7]", this non-convex optimization problem is approximated by reasoning about the trajectory of gradient methods starting from the initialization. Under such circumstances, we employ $\mathcal{W} \in \mathbb{R}^{m \times d}$ as the parameters for the feature extractor, where $m$ denotes the dimension of logits. To formulate the learnable networks, we define $R = C(\mathcal{W}^\top)$, $S = C(X_{id}^\top)$, and $U = C(X_{ood}^\top)$ as the subspaces spanned by the row vectors of the parameterized network, in-domain data, and out-domain data, respectively. Additionally, let $E \in \mathbb{R}^{d \times \dim(R)}$, $F \in \mathbb{R}^{d \times \dim(S)}$, and $G \in \mathbb{R}^{d \times \dim(U)}$ serve as the orthogonal bases for $R$, $S$, and $U$, respectively. Consequently, the algebraic representation of the model and domains can be reformulated linearly as spanning spaces over a set of learnable basis vectors. In this complete inner product space, the following proposition can be claimed as follows:

**Proposition 3.1.** *Model substructures and the feature representations can be effectively corresponded in linear form by the singular value decomposition (SVD) of the feature projections of data into the model space.*

**Discussion (Model):** Define $E^\top F \in \mathbb{R}^{\dim(R) \times \dim(S)}$ as the basis of $C(X_{id}\mathcal{W}^\top)$ spanning the ID (spurious) feature projections. Similarly, $E^\top G \in \mathbb{R}^{\dim(R) \times \dim(U)}$ is the basis of $C(X_{ood}\mathcal{W}^\top)$ spanning the out-domain feature projections. Since the column of $E$ span $R$, we have $\mathcal{W} = Er$ for some $r \in \mathbb{R}^{\dim(R)}$. For every ID instance, the feature projection $r_1 = E^\top Fa$ is used for some $a \in C(E^\top F)$, where $a$ is a column vector of $\mathbb{R}^{\dim(S)}$. Similarly, for every out-domain instance, the feature projection $r_2 = E^\top Gb$ is used for some $b \in C(E^\top G)$, where $b$ is a column vector of $\mathbb{R}^{\dim(U)}$. Therefore, the feature projections of the whole training dataset in the model space

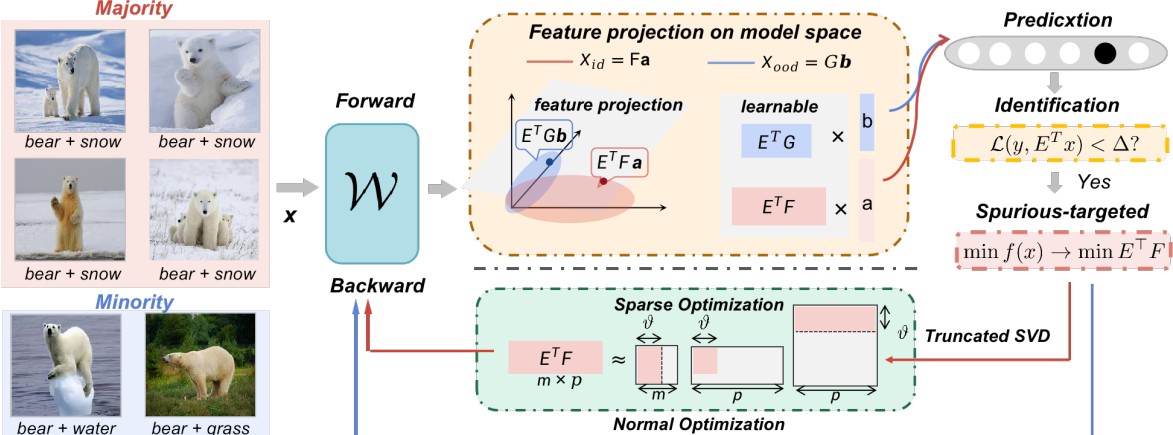

**Figure 1: The training pipeline of SFP.**

can be defined as $r = p_i r_1 + p_o r_2$. Assume $\mathcal{W}^*$ is the optimal set of model parameters, $\mathcal{W}^* = Er^*$, where $r^* = p_i E^\top F a^* + p_o E^\top G b^*$, and $a^*, b^*$ be the true feature projections.

**Discussion (Data):** In $S = C(X_{id}^\top)$ with basis $F$ spanning $X_{id}$, $\forall x_i \in X_{id}, \exists z \in \mathbb{R}^{\dim(S)}, x = (Fz)^\top$. $X_{id} = (FZ)^\top$, where $Z = \{z\}$. Similarly, in $U = C(X_{ood}^\top)$ with basis $G$, $\forall x_{ood} \in X_{ood}, \exists v \in \mathbb{R}^{\dim(U)}, x_{ood} = (Gv)^\top$. $X_{ood} = (GV)^\top$, where $V = \{v\}$.

*3.1.2 Preliminary Optimization Target.*

**Definition 3.2.** Under the OOD setting, applying the same optimization objective to ID data with spurious features and out-domain data without the same spurious features is called undirected learning

**Definition 3.3.** Trained independently from scratch for the same number of iterations, the substructure within the original model having the best OOD generalization performance is defined as the OOD lottery [41].

For the structure-based approach searching the OOD lottery based on undirected learning, the optimization target can be formulated as:

$$\min \ \mathcal{L}(\mathcal{W}, X, Y) = \mathbb{E}_X \|X\mathcal{W} - Y\|_2^2 + \mathcal{S}(\mathcal{W}), \quad (1)$$

where $\mathcal{L}$ is the task-dependent loss function, and $\mathcal{S}$ is the function that induces the sparsity of the model structure to find the target subnetwork. The domain-generalized substructure is described by layer-wise channel saliencies in SFP. To this end, $\mathcal{S}$ is implemented by the squeeze-and-excitation module as suggested in [13]. The value of relevant parameters in $t$-th iteration is represented by subscript $t$, and the optimal value is represented by superscript $*$. Thus, the task loss in $t$-th iteration can be calculated as:

$$\mathcal{L}_t = \|X\mathcal{W}_t - Y\|_2^2 = \|X\mathcal{W}_t - X\mathcal{W}^*\|_2^2, \quad (2)$$

and the gradient is:

$$\frac{\partial \mathcal{L}_t}{\partial \mathcal{W}_t} = 2\left(\mathcal{W}_t - \mathcal{W}^*\right)X^\top X. \quad (3)$$

The orthogonal basis of the model space is regarded as the left singular vectors when performing SVD on the feature projections of data. The right singular vectors correspond to input data features, and the corresponding singular values can be defined as indicators of the importance of the current data features w.r.t. the model structure. To internally observe the impact of ID and out-domain features on the model, the gradient accumulation is further transformed into a linear form:

$$\frac{\partial \mathcal{L}_t}{\partial \mathcal{W}_t} = 2(p_i^2(a_t - a^*)\Sigma_{E^\top F,t}^2 X_{id} + p_o^2(b_t - b^*)\Sigma_{E^\top G,t}^2 X_{ood}), \quad (4)$$

where $\Sigma$ denotes the corresponding singular value matrix, and for simplicity, we omit $t$ under $\Sigma$ in the following discussion. The proof of Eq. Equation (4) is provided in ***Appendix*** A.1.

Since $\dim(U) = q \ll \dim(S) = p$, we have $\min \Sigma_{F^\top G} = \min \Sigma_{G^\top F} = \sigma_{G^\top F}^q$. Similarly, $\min \Sigma_{F^\top E} = \min \Sigma_{E^\top F} = \sigma_{E^\top F}^m$, and $\min \Sigma_{G^\top E} = \min \Sigma_{E^\top G} = \sigma_{E^\top G}^m$. Finally, the model parameters can be calculated as:

$$\mathcal{W}^\infty = \mathcal{W}_0 - 2lr \sum_{t=1}^{\infty} \sum_{i=1}^{m} p_i^2(a_t - a^*)\sigma_{E^\top F,t,i}^2 X_{id}$$
$$- p_o^2(b_t - b^*)\sigma_{E^\top G,t,i}^2 X_{ood}. \quad (5)$$

*3.1.3 Biased Performance on Out-domain and ID Data.* Based on the gradient flow trajectories, we compare the learning process and final performance of the model for spurious and invariant features, respectively. We observe that the model structure obtained by undirected learning clearly differs in performance between ID data and out-of-domain data. With this observation, we propose the following propositions.

**Proposition 3.4.** *Undirected learning (full or sparse training) on biased data distributions can lead to significantly different forward speeds of the model learning along different data feature directions, and the difference has a second-order relationship with the proportion of different data distributions in the training set, i.e.:*

$$\left|\frac{\partial \mathcal{W}_t}{\partial(a_t - a^*)} - \frac{\partial \mathcal{W}_t}{\partial(b_t - b^*)}\right| \approx 2(p_i^2\Sigma_{E^\top F}^2 - p_o^2\Sigma_{E^\top G}^2). \quad (6)$$

**Discussion (Update Gradient):** We compute the direction gradients along the directions of the feature projections of ID and out-domain data, respectively. As shown in Eq. 6, with $p_i \geq p_o$ in the context of OOD, the learning of the basis of the model space is gradually biased towards the directions of spurious features. By

performing SVD on the projection of the basis vector of the feature space through the model space, the obtained singular value matrix can be regarded as the fitting degree of the model on the corresponding data distribution at $t_{th}$ iteration.

**Proposition 3.5.** *Undirected learning (full or sparse training) on biased data distributions causes the model to be more biased towards training features with a larger proportion, bringing about significant performance differences in different data distributions, i.e.:*

$$\mathcal{L}_{ood} - \mathcal{L}_{id} \approx (p_i^2 - p_o^2)(1 - \Sigma_{F^\top G}) + \epsilon > 0, \tag{7}$$

*where $\epsilon$ is the difference of initial feature projections between ID and out-domain data due to model initialization error. The full proof of Eq. 7 is provided in **Appendix A.2** (please download and check).*

Taking the risk difference between ID data and out-domain data of the trained model as the measurement of the OOD generalization, the following conclusion is derived, i.e.,

**Corollary 3.6.** *Undirected learning of networks on highly biased training domains (the dataset consists of a majority data group with spurious features) can only lead to substructures with sub-optimal OOD generalization performance.*

**Discussion (Performance Difference):** The result intuitively shows that the undirectly learned model performs better on feature distributions with larger instance numbers. As shown in Eq. 7, the difference in model performance between out-domain data and ID data is linearly related to the proportion of the corresponding instances and the correlation degree between the different feature distributions. Moreover, when the out-domain data has the same proportion as ID data in the training dataset (i.e., $p_i = p_o$) or the data distributions of them are consistent, the task loss difference between out-domain and ID data can be reduced to zero.

## 3.2 SFP: An Spurious Feature-Targeted Model Pruning Method

To address the problem of sub-optimal OOD substructure caused by undirected training, we propose a novel method to effectively remove model branches that are only strongly correlated with spurious features. As demonstrated in Fig. 1, the pipeline consists of two stages, including spurious feature identification and model sparse training. Specifically, SFP identifies large spurious feature components within ID instances with high probability by observing the loss during training. It then can perform spurious feature-targeted model sparsity by analyzing the SVD of the feature projection matrix between the data and model space. We also provide a detailed theoretical analysis of both stages of the proposed SFP in the following part.

*3.2.1 Spurious Feature Identification.* As shown in Proposition.3.5, if no intervention is applied, a model trained on a highly biased data distribution can be gradually biased towards ID data with lower prediction loss. Since the loss difference between ID and out-domain data can be approximately computed by $(p_i^2 - p_o^2)(1 - \sigma_{F^\top G})$, it is, therefore, can be adopted as the identification criterion for spurious features in each iteration. In brief, if the loss corresponding to the current data is lower than a threshold $\Delta$, then the current data is likely to be an ID instance dominated by spurious features. Then

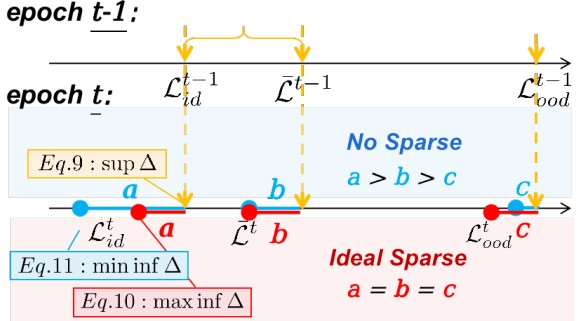

**Figure 2: Identification of the ID instances dominated by spurious features. At epoch t, if no intervention is applied, the average loss drop on all data (blue b) should be smaller than that on ID data (blue a) and larger than that on out-domain data (blue b). The red line denotes an ideal regularization effect: the loss drops uniformly on all data.**

we can further prune the spanning sets of model space along the directions of these spurious feature projections. To compute $\Delta$, we first investigate the average loss in the $t - 1$-th iteration as:

$$\bar{\mathcal{L}}^{t-1} \approx \mathcal{L}_{id}^{t-1} + p_o(p_i - p_o)(1 - \sigma_{F^\top G}^{t-1}). \tag{8}$$

As shown in Fig. 2, since $\mathcal{L}_{id}^t < \mathcal{L}_{id}^{t-1}$, we have:

$$\sup \mathcal{L}_{id}^t = |\bar{\mathcal{L}}^{t-1} - p_o(p_i - p_o)(1 - \sigma_{F^\top G}^{t-1})|. \tag{9}$$

Similar with Eq. 8, the lower bound of the loss on ID data at $t$-th iteration can be computed as:

$$\inf \mathcal{L}_{id}^t = |\bar{\mathcal{L}}^t - p_o(p_i - p_o)(1 - \sigma_{F^\top G}^t)|. \tag{10}$$

The spurious feature-targeted regularization forces the model to learn invariant features and achieve fair loss reduction on all instances: $|\mathcal{L}_{id}^t - \mathcal{L}_{id}^{t-1}| = |\bar{\mathcal{L}}^t - \bar{\mathcal{L}}^{t-1}|$. Therefore, the ideal lower bound of the ID loss at $t$-th iteration is:

$$\begin{aligned} \inf \mathcal{L}_{id}^t = &|\bar{\mathcal{L}}^{t-1} - p_o(p_i - p_o)(1 - \sigma_{F^\top G}^{t-1})| \\ &- |\bar{\mathcal{L}}^t - \mathcal{L}^{\bar{t}-1}|. \end{aligned} \tag{11}$$

Thus, $\mathcal{L}_{id}^t$ is highly likely to be located in the range of $[\min \inf \mathcal{L}_{id}^t, \sup \mathcal{L}_{id}^t]$. The upper bound is used to compute $\Delta$ for identifying instances dominated by spurious features.

*3.2.2 Spurious Feature-Targeted Pruning.* SFP reacts to spurious feature-related instances by weakening their corresponding spurious feature projections into the model space, which can prevent the model from over-fitting on identified spurious features. To analyze the projections from data into the model space, we define $\Xi \in \mathbb{R}^{m \times m}$, $\Lambda \in \mathbb{R}^{p \times p}$, and $\Gamma \in \mathbb{R}^{q \times q}$ as the normalized orthogonal basis of $C(E^\top E)$, $C(E^\top F)$, and $C(E^\top G)$, spanning the optimal model projections, the feature projections of ID data into the model space, and the feature projections of out-domain data into the model space, respectively. $\xi_i$, $\lambda_i$, and $\gamma_i$ denote the $i$-th column vectors in $\Xi$, $\Lambda$, and $\Gamma$, respectively. The following lemma illustrates the effectiveness of SFP, and its proof can be found in **Appendix** A.3.

**Lemma 3.7.** *Spurious feature-targeted model sparsity can effectively reduce the performance deviation of the learned model between in-domain data and out-domain data:*

$$\left| \frac{\mathcal{R}(X_{ood}) - \mathcal{R}(X_{id})}{\mathcal{R}(X_{ood})^{sparse} - \mathcal{R}(X_{id})^{sparse}} \right|$$

$$\approx \left| \frac{\sum_{j=1}^{m} p_o \tilde{\sigma}_j \xi_j \gamma_j X_{ood} - \sum_{i=1}^{m} p_i \sigma_i \xi_i \lambda_i X_{id}}{\sum_{j=1}^{m} p_o \tilde{\sigma}_j \xi_j \gamma_j X_{ood} - \sum_{i=1}^{\vartheta} p_i \sigma_i \xi_i \lambda_i X_{id}} \right| \geq 1, \quad (12)$$

*where $\mathcal{R}(\cdot)$ is the empirical risk function. $\sigma_i$ and $\tilde{\sigma}_i$ is the i-th maximum in $\Sigma_{E^\top F}$ and $\Sigma_{E^\top G}$, and we have $\sigma > 0$ since the singular values are non-negative. m and $\vartheta$ are the rank of the singular value matrix after performing compact SVD and truncated SVD on the projections, respectively.*

PROOF OF LEMMA.3.7. As mentioned earlier, the projection space before the model sparsity can be represented as:

$$Er = \sum_{i=1}^{m} \left( p_i \sigma_i \xi_i \lambda_i^\top + p_o \tilde{\sigma}_i \xi_i \gamma_i^\top \right). \quad (13)$$

Specifically, SFP first performs SVD on the feature projections, which maps input data to a set of coordinates based on the orthogonal basis of model space. The matrices of left and right singular vectors correspond to the standard orthogonal basis of the model space and data space, respectively. The matrix of singular values corresponds to the direction weight of the action vectors in the projection matrix. SFP prunes the model by trimming the smallest singular values in $\Sigma$ as well as their corresponding left and right singular vectors. In this way, SFP can remove the spurious features in ID data space and substructures in the model space simultaneously in a spurious feature-targeted manner along the directions with weaker actions for projection. Then, the projection space with only the most important $\vartheta$ singular values can be formalized as:

$$Er^{sparse} = p_i \Xi \Sigma_{E^\top F} \Lambda^{-1} + p_o \xi \Sigma_{E^\top G} \Gamma^{-1}$$

$$= \sum_{i=1}^{\vartheta} p_i \sigma_i \xi_i \lambda_i^\top + \sum_{j=1}^{m} p_o \tilde{\sigma}_j \xi_j \gamma_j^\top. \quad (14)$$

Based on the representation of the projection spaces, the model response to data features $\mathcal{R}(X) = ErX$ can be calculated as:

$$\mathcal{R}(X) = \left\{ p_i \Xi \Sigma_{E^\top F} \Lambda^{-1} + p_o \xi \Sigma_{E^\top G} \Gamma^{-1} \right\}^\top X^\top$$

$$= \sum_{i=1}^{m} \left\{ p_i \sigma_i \xi_i \lambda_i^\top X^\top + p_o \tilde{\sigma}_i \xi_i \gamma_i^\top X^\top \right\}. \quad (15)$$

$\square$

## 3.3 Correspondence between Model Substructure and Spurious Features

In this section, we theoretically demonstrate that, with a reasonable setting of the sparse penalty for ID data, SFP can effectively reduce the overfitting of the model on spurious features while retaining the learning on invariant features. Specifically, we define $f^l(x)$ as the feature maps output of $x$ at layer $l$. It represents the projection of $x$ onto the model space defined over the spanning set $E$ to be learned. We abbreviate the final probabilities as $f(x)$ for simplification. Referring to Sec. 3.2.1, we have $x \in X_{id}$ if $\mathcal{L}_{ce}(x) \leq \Delta$. Thus,

the optimization target of SFP can be formulated as:

$$\min_E \mathbb{E}_{x \sim X} \mathcal{L}_{ce}(x, \mathcal{W}) + \eta \sum_{l=1}^{L} \mathbb{E}_{x \sim X_{id}} ||f^l(x)||_1, \quad (16)$$

where $\eta$ is the sparsity factor imposed on the feature projections for the identified ID data. Lemma 3.8 elucidates the setting of $\eta$. For a detailed proof, please refer to ***Appendix*** A.4.

**Lemma 3.8.** *Define $e = |f^*(x) - f(x)|$ as the $l_1$-norm between the groudtruth $f^*(x)$ and $f(x)$. When $\eta < 2e$, SFP can effectively reduce the learning of the model towards spurious features while keeping the performance on the other features.*

PROOF OF LEMMA.3.8: The prediction errors of feature projections $L_f$ can be defined as:

$$L_f = |f^*(x) - f(x)|^2$$

$$= \sum_{i,j=j_1 \cup j_2} (f^*(x) - \sigma_{i,j_1} \xi_i^\top \lambda_{j_1} - \sigma_{i,j_2} \xi_i^\top \gamma_{j_2})^2, \quad (17)$$

and the corresponding gradient is:

$$\frac{\partial L_f}{\partial \sigma_{i,j_1} \xi_i} = \frac{\partial e^2}{\partial \sigma_{i,j} \xi_i} = 2e \frac{\partial e}{\partial \sigma_{i,j} \xi_i}$$

$$= 2e \frac{\left| f^*(x) - \sigma_{i,j_1} \xi_i^\top \lambda_{j_1} - \sigma_{i,j_2} \xi_i^\top \gamma_{j_2} \right|}{\partial \sigma_{i,j} \xi_i} = -2e \lambda_{j_1}, \quad (18)$$

where $i$ and $j$ are the index of column vectors in the orthogonal basis for model space and feature space, respectively. For out-domain data, the gradient of the column vectors in the OOD projection matrix interacting with the $j_{th}$ feature vector is $-2e\gamma_{j_2}$. Then, split the in-domain features into spurious features $F'$ and invariant features $IN$ and out-domain features into unknown features $G'$ and invariant features $IN$. With a high probability under the OOD setting, we assume $F'$ and $G'$ are orthogonal. To achieve the spurious feature-targeted unlearning and invariant feature-targeted learning of the model, we need to satisfy the following constraint:

$$2ep_i\lambda_{IN} + 2ep_o\gamma_{IN} - p_i\eta\lambda_{IN} > 2ep_o\gamma_{G'}$$

$$\Rightarrow \eta \leq \frac{2ep_i\lambda_{IN} + 2ep_o\gamma_{IN} - 2ep_o\gamma_{G'}}{p_i\lambda_{IN}} \approx 2e. \quad (19)$$

$\square$

Since the de-learning rate of the spurious feature is positively correlated with $\eta$, the upper bound $\eta = 2e$ is taken.

## 4 EXPERIMENTS

In this section, we conducted extensive experiments on the DomainBed benchmarks [10] and other datasets that are widely used in the latest OOD studies. Due to space constraints, some experimental details are provided in the ***Appendix*** B and C.

## 4.1 Experimental Setting

**Datasets and Procedure.** The proposed method is initially evaluated within the DomainBed framework using four datasets: ColoredMNIST (**CMNIST**), RotatedMNIST (**RMNIST**), as well as the multi-domain image classification datasets **PACS**, **OfficeHome**, **TerraInc**, and **DomainNet** [10]. To ensure comprehensive benchmarking, three synthetic datasets — FullColoredMNIST (**FCMNIST**),

**ColoredObject**, and **SceneObject** — are included, along with two real-world image datasets, **CelebA** [24] and **WaterBirds** [35]. Figure 1 illustrates the three synthetic datasets not encompassed within the DomainBed, and more details are provided in **_Appendix_** C.1.

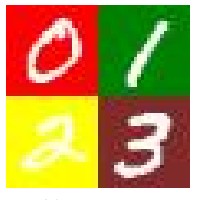 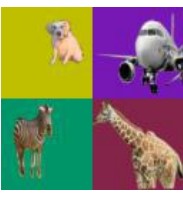 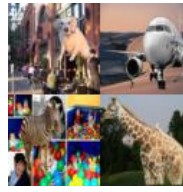

(a) FCMNIST  (b) ColoredObject  (c) SceneObject

Figure 3: Visualization of three synthetic OOD datasets.

**Model and Implementation.** To ensure a robust and equitable evaluation, the experimental settings in this work are consistent with the common practice established in antecedent studies. Specifically, for Rotated, Colored, and FCMNIST datasets, we use the 4-layer 3x3 ConvNet architecture as introduced in DomainBed. For the VLCS and PACS datasets, we utilize the ResNet-18 architecture as in IIB [18], with the default hyperparameters set in DomainBed. Additionally, for other larger datasets, we adopt the ResNet-50 architecture following the experimental settings outlined by previous works [30, 31]. All experiments are conducted on a workstation equipped with 8 Nvidia GTX 3090TI GPUs and a 3.6-GHZ Intel Core i9-9900KF CPU. The learning rate is initialized at 0.001 for digit datasets and 0.01 for object datasets. We employ the Adam optimizer for optimization in relatively simple image datasets, while SGD for more complex ones.

## 4.2 Comparison on DomainBed Benchmark

The experiment results on DomainBed demonstrate the superior performance of SFP over the state-of-the-art approaches. As shown in Table 1, SFP achieves the highest average accuracy of 72.8%, outperforming the benchmarked ERM (which is meticulously tuned within DomainBed and serves as a robust baseline) by 2.2%. On

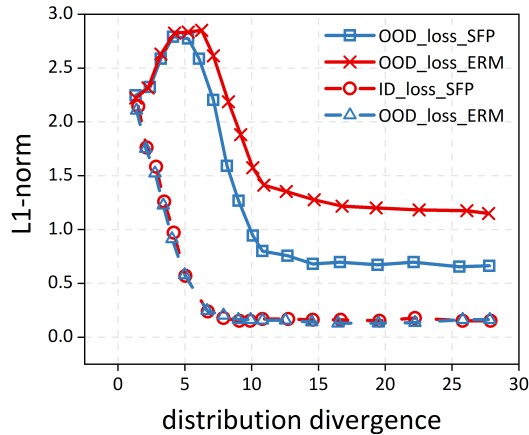

Figure 4: Training loss visualization

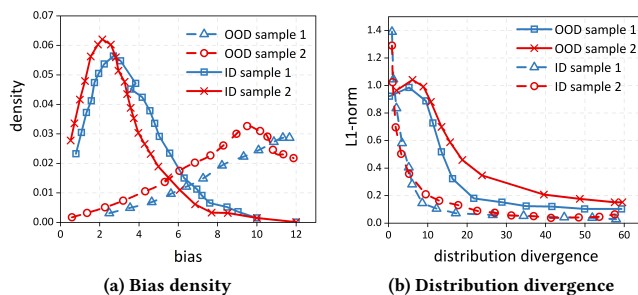

(a) Bias density  (b) Distribution divergence

Figure 5: The probability density of bias between the max value and the others in predicted distribution.

smaller datasets such as Colored and Rotated MNIST, most methods exhibit limited effectiveness. In contrast, SFP stands out by achieving an accuracy improvement of up to 14.0%, highlighting its robust feature-based recognition and suppression capabilities against correlation shifts. On larger datasets, SFP maintains satisfactory performance, demonstrating a remarkable accuracy increase up to 2.9% and 9.4% on VLCS and PACS, respectively. Notably, on the OfficeHome dataset, SFP boosts the OOD accuracy from 68.6% to 71.8%. The results also underscore the disadvantages of SOTAS in effectively addressing the correlation and diversity shifts simultaneously. For instance, while the ARM method excels in mitigating correlation shifts on Colored MNIST, it falters when confronted with diversity shifts in the OfficeHome dataset. Conversely, IIB performs well in scenarios involving diversity shifts but exhibits mediocre performance in correlation shift scenarios. Differently, SFP exhibits superior performance in most cases, emerging as a leading approach in the field of OOD generalization. More experimental details are provided in **_Appendix_** C.2.

## 4.3 Comparison on Other Benchmarks

We also conduct experiments on several widely-used datasets not included in DomainBed. For synthetic FCMNIST and ColoredObject datasets, bias coefficients (indicating the extent of data shift) are set as (0.8, 0.6, 0.0). This implies that the digits in the two training domains are spuriously colored with probabilities of 0.8 and 0.6, while images in the test domain are randomly colored. For SceneObject dataset, we set the biased ratios as (0.9, 0.7, 0.0), futher hampering the model's capture of invariant features.

We compare SFP with the most comparable MRM, as well as their combined variants with IRM [2], V-REx [15], and DRO [31], on three synthetic datasets including FCMNIST, ColoredObject, and SceneObject. The results are shown in Table 2, demonstrating the superior performance of SFP under both independent and combined modes. To be specific, the results show that MRM compromises the generalization performance of the original algorithm in some cases. For example, the DRO algorithm independently achieves a test accuracy of 31.31% on SceneObject. However, when combined with MRM, the performance drops to 29.38%, while SFP contributes to an increased accuracy of 31.78%.

We also compare SFP with state-of-the-art SparseIRM [45] on FCMNIST with two different architectures, i.e., ResNet18 and MLP.

Table 1: DomainBed benchmark: Performance comparison (Accuracy %) between the proposed SFP method and the state-of-the-art domain generalization methods. "-" represents the missing data due to partially different settings. "Average" reports the average accuracy over all the datasets. The best accuracy in each case is in boldface.

| Algorithm | CMNIST | RMNIST | VLCS | PACS | OfficeHome | TerraInc | DomainNet | Average |
|---|---|---|---|---|---|---|---|---|
| | MLP | MLP | ResNet-18 | ResNet-18 | ResNet-50 | ResNet-50 | ResNet-50 | |
| ERM [33] | $57.8_{\pm0.2}$ | $97.8_{\pm0.1}$ | $77.2_{\pm0.4}$ | $83.0_{\pm0.7}$ | $66.4_{\pm0.5}$ | $53.0_{\pm0.3}$ | $41.3_{\pm0.1}$ | 70.6 |
| IRM [2] | $67.7_{\pm1.2}$ | $97.5_{\pm0.2}$ | $76.3_{\pm0.6}$ | $81.5_{\pm0.8}$ | $63.0_{\pm2.7}$ | $50.5_{\pm0.7}$ | $28.0_{\pm5.1}$ | 69.0 |
| GroupDRO [31] | $61.1_{\pm0.9}$ | $97.9_{\pm0.1}$ | $77.9_{\pm0.5}$ | $83.5_{\pm0.2}$ | $66.2_{\pm0.6}$ | $52.4_{\pm0.1}$ | $33.4_{\pm0.3}$ | 67.5 |
| Mixup [39] | $58.4_{\pm0.2}$ | $98.0_{\pm0.1}$ | $77.7_{\pm0.6}$ | $83.2_{\pm0.4}$ | $68.0_{\pm0.2}$ | $54.4_{\pm0.3}$ | $39.6_{\pm0.1}$ | 63.3 |
| MLDG [19] | $58.2_{\pm0.4}$ | $97.8_{\pm0.1}$ | $77.2_{\pm0.9}$ | $82.9_{\pm1.7}$ | $66.6_{\pm0.3}$ | $52.0_{\pm0.1}$ | $41.6_{\pm0.1}$ | 68.0 |
| MMD [1] | $63.3_{\pm1.3}$ | $98.0_{\pm0.1}$ | $77.3_{\pm0.5}$ | $83.2_{\pm0.2}$ | $66.2_{\pm0.3}$ | $50.0_{\pm0.4}$ | $23.5_{\pm9.4}$ | 66.2 |
| CDANN [21] | $59.5_{\pm2.0}$ | $97.9_{\pm0.0}$ | $77.5_{\pm0.2}$ | $78.8_{\pm2.2}$ | $65.3_{\pm0.5}$ | $50.8_{\pm0.6}$ | $38.5_{\pm0.2}$ | 66.9 |
| MTL [3] | $57.6_{\pm0.3}$ | $97.9_{\pm0.1}$ | $76.6_{\pm0.5}$ | $83.7_{\pm0.4}$ | $66.5_{\pm0.4}$ | $52.2_{\pm0.4}$ | $40.8_{\pm0.1}$ | 67.9 |
| SagNet [27] | $58.2_{\pm0.3}$ | $97.9_{\pm0.0}$ | $77.5_{\pm0.3}$ | $82.3_{\pm0.1}$ | $67.5_{\pm0.2}$ | $52.5_{\pm0.4}$ | $40.8_{\pm0.2}$ | 68.1 |
| ARM [42] | $63.2_{\pm0.7}$ | $98.1_{\pm0.1}$ | $76.6_{\pm0.5}$ | $81.7_{\pm0.2}$ | $64.8_{\pm0.4}$ | $51.2_{\pm0.5}$ | $36.0_{\pm0.2}$ | 67.4 |
| V-REx [15] | $67.0_{\pm1.3}$ | $97.9_{\pm0.1}$ | $76.7_{\pm1.0}$ | $81.3_{\pm0.9}$ | $65.7_{\pm0.3}$ | $51.4_{\pm0.5}$ | $30.1_{\pm3.7}$ | 67.2 |
| RSC [14] | $58.5_{\pm0.5}$ | $97.6_{\pm0.1}$ | $77.5_{\pm0.5}$ | $82.6_{\pm0.7}$ | $66.5_{\pm0.6}$ | $52.1_{\pm0.2}$ | $38.9_{\pm0.6}$ | 67.7 |
| IIB [18] | - | - | $77.2_{\pm1.6}$ | $83.9_{\pm0.2}$ | $68.6_{\pm0.1}$ | $57.4_{\pm0.7}$ | $41.5_{\pm2.3}$ | - |
| Fishr [30] | $68.8_{\pm1.4}$ | $97.8 \pm 0.1$ | - | - | $68.2_{\pm0.2}$ | $53.6_{\pm0.4}$ | $41.8_{\pm0.2}$ | - |
| SDL [40] | $58.8_{\pm2.2}$ | - | - | $84.8_{\pm0.6}$ | $63.9_{\pm0.1}$ | - | - | - |
| **SFP** | $71.6_{\pm0.3}$ | $98.3_{\pm1.4}$ | $79.2_{\pm0.7}$ | $90.7_{\pm0.1}$ | $71.8_{\pm0.1}$ | $57.8_{\pm0.3}$ | $40.0_{\pm0.7}$ | **72.8** |

Table 2: OOD generalization performance on FullColoredMNIST, ColoredObject, and SceneObject. "MRM+X" and "SFP+X" indicate the integration of MRM/SFP in the "X" algorithm. The "Unbiased" row reports the original accuracy for each dataset without data distribution shifts.

| Method | FCMNIST | ColoredObject | SceneObject |
|---|---|---|---|
| ERM | 62.2 | 59.2 | 27.4 |
| MRM | 81.0 | 60.7 | 26.7 |
| **SFP** | **84.3** | **61.01** | **28.4** |
| IRM | 78.0 | 62.9 | 36.9 |
| MRM +IRM | 89.3 | 64.5 | 36.9 |
| **SFP+IRM** | **89.9** | **65.8** | **38.1** |
| V-REx | 87.8 | 64.7 | 36.7 |
| MRM +V-REx | 92.2 | 64.5 | 36.7 |
| **SFP+V-REx** | **93.4** | **66.1** | **37.9** |
| DRO | 62.9 | 66.8 | 31.3 |
| MRM +DRO | 80.5 | 66.2 | 29.4 |
| **SFP+DRO** | **85.2** | **68.4** | **31.8** |
| UNBIASED | 94.0 | 75.8 | 45.5 |

Specifically, SFP outperforms SparseIRM with 3.41% higher test accuracy on MLP and even 29.12% on ResNet18. An interesting phenomenon is that, on small MLP, SparseIRM exhibits an obvious two-stage trend, which is consistent with regular non-feature-targeted model pruning. Differently, SFP consistently shows a stable learning process and achieves higher performance in both ID (train) and OOD (test) environments. Due to space constraints, the experimental details are provided in ***Appendix*** C.3.

### 4.4 Ablation Study

**Loss tracking.** We visualize and compare loss values between ERM and our proposed SFP to assess the efficacy of our introduced regularization term. As shown in Fig.4, throughout the training process, the loss of in-domain (ID) instances consistently remains lower than that of out-domain instances, validating Proposition 3.5. In ERM, the rapid convergence of ID instance loss (depicted by red lines) indicates an excessive focus on biased data, leading to overfitting spurious features and neglecting invariant features. Conversely, in SFP, the gap between loss values for ID and out-domain instances narrows significantly, underscoring the effectiveness of spurious feature-targeted pruning. What's more, the optimization of SFP won't hinder convergence speed as well as adversely affects the performance of ID instances.

**Prediction confidence.** The inherent motivation of SFP originates from scrutinizing the behavioral disparities between ID samples and OOD samples under ERM, which is illustrated via two empirical experiments as follows. We first measure the bias between the maximum value and other values in the logits vectors corresponding to different samples, where the maximum typically

represents the prediction. A large logits discrepancy suggests a significant divergence between the probability densities of the predicted class and others, which can be used as a metric for gauging the prediction confidence. The results, depicted in Fig. 5a, reveal that ID samples generally exhibit larger logits discrepancies compared to OOD samples, indicating a tendency of the current model to allocate greater confidence to the predictions of ID samples.

Additionally, we evaluate the $l_1$-norm between the predicted and true distributions over different classes to gauge the degree of the model capturing different features. The results are shown in Fig. 5b. It's evident that the distribution loss of ID samples sharply decreases in the early training stages but gradually slows down afterward. Conversely, OOD samples initially show a slight increase in distribution loss, followed by a steep decrease. This early training behavior suggests that the model initially prioritizes spurious correlations, but as training progresses, SFP mitigates the fit of spurious correlations while promoting the learning of invariant features. As a result, the downward trend of distribution loss for ID samples decelerates, while the trend for OOD samples starts to rise.

**Sparsity analysis.** Prior structure-based OOD studies usually utilize human-crafted hyperparameters to find a suitable functional OOD substructure. In contrast, our method treats the sparsity coefficients $(\Delta, \eta)$ as dynamic variables that are calculated dynamically during training, i.e., *the proposed SFP intelligently determines the optimal OOD sparsity and structure based on inherent data attributes.* Specifically, $(\Delta)$ gives a sparsity threshold based on inherent statistical and geometric biases within the data (e.g., Eq. 9-11), and $\eta$ adjusts the penalty strength based on dynamic training feedback (e.g., Eq. 6). To empirically evaluate the sparsity of our model and, at the same time, provide a quantitative impact of $\eta$ on OOD accuracy, we conduct experiments on varied offsets to the theoretically computed $\eta$ (2e). Specifically, the offsets are ranged in [-1.0, -0.5, 0.0, 0.5, 1.0]. The results regarding model sparsity and test accuracy are shown in Fig. 6. The corresponding OOD accuracy are [73.01262%, 79.84853%, 86.30715%, 84.19074%, 76.23703%], and the pruning rates are ranged in [27.94951%, 45.09116%, 56.70407%, 62.09122%, 74.40112%]. The results demonstrate that the autonomous acquisition of sparsity and sparse structures (offset of 0) yields superior OOD performance than empirical sparse settings.

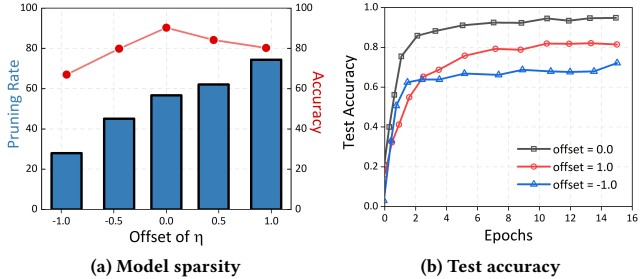

(a) Model sparsity       (b) Test accuracy

**Figure 6: The effect of different $\eta$ values on the model sparsity and accuracy.**

**Feature visualization.** To explore the SFP model's learned representations, we visualize the extracted features using t-distributed

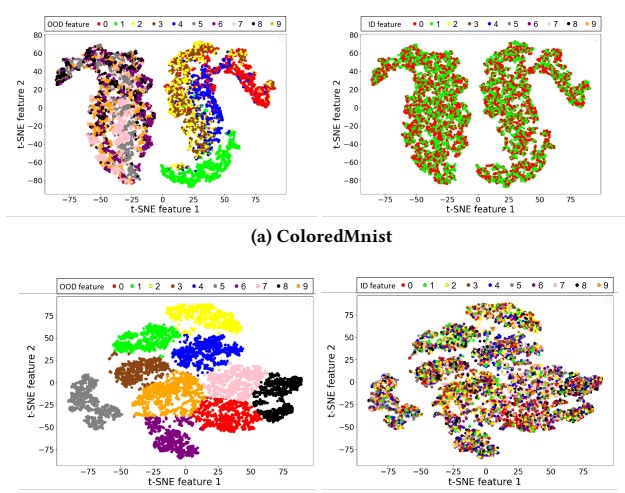

(a) ColoredMnist

(b) FullColoredMnist

**Figure 7: The visualization of the features learned by SFP.**

stochastic neighbor embedding (t-SNE) for dimensionality reduction. Experiments are conducted on FullColoredMnist and ColoredMnist datasets, tailored for ten-class and binary classification tasks, respectively. The former integers 0 to 9 are ten classes with one-to-one associated colors. The latter integers 0 to 4 are labeled as 0 (red) and integers 5 to 9 as 1 (green). The models are trained on domain-related samples and tested on domain-unrelated samples with random colors.

The visualization is shown in Fig. 7, where each data point represents an image. Notably, the spatial arrangement corresponds to the reduced shape features. The features cluster into two groups for ColoredMnist and ten groups for FullColored MNIST. All left subplots color each point based on invariant features, i.e., samples with the same digit are colored identically. For example, as shown in Fig. 4a, each cluster contains points belonging to class 0 (digits 0-4) or class 1 (digits 5-9). Conversely, all right subplots color each point based on spurious features, where samples with the same color foreground or background (e.g., red 2 and red 3) are colored identically. The results are shown in Figures 4b and 5b, each cluster (class) involves diverse spurious feature. This indicates that clustered features are specific to invariant digit shapes and remain unaffected by color variations, demonstrating SFP could successfully acquire disentangled representations.

## 5 CONCLUSION

In this paper, we introduce a novel spurious feature-targeted model pruning framework, dubbed SFP, designed to automatically explore the optimal model substructure for improved out-of-distribution (OOD) generalization. By effectively identifying spurious features within in-distribution (ID) instances during training, SFP can selectively remove model branches that heavily depend on these spurious features. As a result, SFP attenuates the impact of spurious features on the model's representation space and guides the model learning process toward invariant features. Additionally, we

provide a detailed theoretical analysis to establish the rationality of our approach and offer a proof framework for understanding OOD structures via model sparsity. Experimental results corroborate the effectiveness of our proposed method.

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

# A  PRELIMINARIES AND PROOFS

## A.1  Preliminaries on Important Notations

We first define the following set of symbols:

| Data | Row Space | Basis | Coordinate vector | Singular value matrix |
|------|-----------|-------|-------------------|------------------------|
| $X_{id} \in \mathbb{R}^{p \times d}$ | $S \in \mathbb{R}^d, \dim(S) = p$ | $F \in \mathbb{R}^{d \times p}$ | $z \in \mathbb{R}^p$ | $\mathbf{I}_{F^\top F}$ |
| $X_{ood} \in \mathbb{R}^{q \times d}$ | $U \in \mathbb{R}^d, \dim(U) = q$ | $G \in \mathbb{R}^{d \times q}$ | $v \in \mathbb{R}^p$ | $\mathbf{I}_{G^\top G}$ |
| $\mathcal{W} \in \mathbb{R}^{m \times d}$ | $R \in \mathbb{R}^d, \dim(R) = m$ | $E \in \mathbb{R}^{d \times m}$ | $r \in \mathbb{R}^m$ | $\mathbf{I}_{E^\top E}$ |
| $f(X_{id}) \in \mathbb{R}^{p \times m}$ | $A \in \mathbb{R}^m, \dim(A) = p$ | $E^\top F \in \mathbb{R}^{m \times p}$ | $a \in \mathbb{R}^p$ | $\Sigma_{E^\top F}$ |
| $f(X_{ood}) \in \mathbb{R}^{q \times m}$ | $B \in \mathbb{R}^m, \dim(B) = q$ | $E^\top G \in \mathbb{R}^{m \times q}$ | $b \in \mathbb{R}^q$ | $\Sigma_{E^\top G}$ |

wherein $p$ and $q$ are the sample numbers of the in-distribution data and out-of-distribution data in the training set, respectively. $d$ and $m$ are the dimensions of the data and the model, respectively. Based on the above table, for each sample $x_{id} \in X_{id}$, it can be denoted as $x_{id} = (Fz)^\top$. Additionally, $X_{id}$ is represented for the in-distribution training set as $(Fz)^\top$, where $z$ stands for the set of $z$. Similarly, $v$, $r$, $a$ and $b$ are the set of $v$, $r$, $a$ and $b$, respectively.

**Definition A.1** (Row space). Give a matrix $\mathcal{W} \in \mathbb{R}^{m \times d}$, the rowspace $R$ of $\mathcal{W}$ is the span of the row vectors of $\mathcal{W}$, which can be denoted as $C(\mathcal{W}^\top)$.

**Definition A.2** (Basis and Coordinate). Let $E \in \mathbb{R}^{d \times \dim(R)}$ have orthonormal columns that span $R$. For arbitrary vector $w \in \mathcal{W}$, there exist a $r \in \mathbb{R}^{\dim(R)}$ that satisfies $w = Er$, wherein $E$ is called the orthonormal basis of space $R$ and $r$ is the coordinate vector of $w$ under $E$.

PROOF OF EQUATION. 4.

$$\sum_{t=1}^\infty \frac{\partial \mathcal{L}_t}{\partial \mathcal{W}_t} = \sum_{t=1}^\infty 2 \left(\mathcal{W}_t - \mathcal{W}^*\right) \mathbb{E}[X^\top X]$$

$$= 2 \sum_{t=1}^\infty (Er_t - Er^*)^\top \left\{ p_{id} Fz(Fz)^\top + p_{ood} Gv(Gv)^\top \right\}$$

$$= 2 \sum_{t=1}^\infty \left\{ EE^\top (p_{id} Fa_t^\top + p_{ood} Gb_t^\top) - Er^* \right\}^\top \times \left\{ p_{id} Fz(Fz)^\top + p_{ood} Gv(Gv)^\top \right\}$$

$$= 2 \sum_{t=1}^\infty \left\{ (p_{id} a_t F^\top + p_{ood} b_t G^\top) EE^\top - r^{*\top} E^\top \right\} \times \left\{ p_{id} Fz(Fz)^\top + p_{ood} Gv(Gv)^\top \right\} \tag{20}$$

$$= 2 \sum_{t=1}^\infty \left( p_{id} a_t F^\top + p_{ood} b_t G^\top \right) EE^\top \left\{ p_{id} Fz(Fz)^\top + p_{ood} Gv(Gv)^\top \right\} - r^{*\top} E^\top \mathbb{E}[X^\top X]$$

$$= 2 \sum_{t=1}^\infty \left\{ p_i^2 a_t F^\top EE^\top Fz(Fz)^\top + p_o^2 b_t G^\top EE^\top Gv(Gv)^\top \right\}$$

$$+ p_i p_o a_t F^\top EE^\top Gv(Gv)^\top + p_i p_o b_t G^\top EE^\top Fz(Fz)^\top - r^{*\top} E^\top \mathbb{E}[X^\top X]$$

Given that $E^\top F$ and $E^\top G$ represent the feature projections of the ID data basis and OOD data basis in the model space, respectively, it follows that $F^\top EE^\top G \ll F^\top EE^\top F$ and $F^\top EE^\top G \ll G^\top EE^\top G$. Consequently, Equation 20 can be further simplified as:

$$\sum_{t=1}^\infty \frac{\partial \mathcal{L}_t}{\partial \mathcal{W}_t} \approx 2 \sum_{t=1}^\infty p_i^2 a_t F^\top EE^\top Fz(Fz)^\top + p_o^2 b_t G^\top EE^\top Gv(Gv)^\top - r^{*\top} E^\top \mathbb{E}[X^\top X]$$

$$\approx 2 \sum_{t=1}^\infty \left\{ p_i^2 \widetilde{a_t} \Sigma_{E^\top F,t}^2 X_{id} + p_o^2 \widetilde{b_t} \Sigma_{E^\top G,t}^2 X_{ood} \right\} \tag{21}$$

where $\widetilde{a_t} = a_t - a^*$ and $\widetilde{b_t} = b_t - b^*$. Note that in order to make the expression clearer, we omit the representation of some coordinate vectors ($z$) in Eq. 2, so as to highlight the transformation represented by singular matrix $\Sigma_{E^\top F,t}$ and $\Sigma_{E^\top G,t}$. □

## A.2 Proofs For Biased Performance on OOD and ID Data

**Definition A.3 (Projector).** Give a subspace $S$ of $\mathbb{R}^d$, and P is the projection matrix which projects a vector $x \in \mathbb{R}^d$ into the subspace $S$. If subspace $S$ has a orthonormal basis $E$, we have:

$$P^2 = P^\top = P$$
$$P(x) = E^\top x \tag{22}$$

**Lemma A.4.** *There exists householder matrix $H = I - 2uu^H$ satisfying $det(H) = -1$.*

**Lemma A.5.** *If $A^*$ is the conjugate transpose of A, then $A^*$ has the same nonzero singular values with A.*

PROOF OF LEMMA A.5. Given $A \in C^{m \times n}$ and $A$ has the rank of $r(A) = \min(m, n)$, $A^* \in C^{n \times m}$. Let $A = C_r^{m \times n}$ Then we have $A^*A$ and $AA^*$ are both non-negative definite Hermite matrices. It can be obtained for Lemma A.4 that, for all $\lambda \in \mathbb{R}$, we have:

$$\lambda^m \left| I_n - AA^* \right| = \lambda^n \left| I_m - A^*A \right| \tag{23}$$

$\square$

PROOF OF EQUATION. 7.

$$\mathcal{L}_{ood} = (W_\infty - W^*)X_{ood}^\top$$
$$= \left\{ W_0 - 2lr \lim_{t \to \infty} \sum_{t=1}^\infty \frac{\partial \mathcal{L}_t}{\partial W_t} - W^* \right\} X_{ood}^\top$$
$$= \epsilon_{ood} - 2lr \sum_{t=1}^\infty \left\{ p_i^2 \Sigma_{E^\top F,t}^2 X_{id} + p_o^2 \Sigma_{E^\top G,t}^2 X_{ood} \right\} X_{ood}^T \tag{24}$$
$$\approx \epsilon_{ood} - 2lr \sum_{t=1}^\infty p_i^2 \Sigma_{E^\top F,t}^2 \Sigma_{F^\top G} + p_o^2 \Sigma_{E^\top G,t}^2 I_{G^\top G}$$
$$\mathcal{L}_{id} = \left( W_\infty - W^* \right) X_{id}^\top$$
$$\approx \epsilon_{id} - 2lr \sum_{t=1}^\infty p_i^2 \Sigma_{E^\top F,t}^2 I_{F^\top F} + p_o^2 \Sigma_{E^\top G,t}^2 \Sigma_{G^\top F} \tag{25}$$

From Lemma. 2, we have $\Sigma_{G^\top F} = \Sigma_{F^\top G}$. And since $\dim(U) = q \ll \dim(S) = p$, the smallest singular value in singular value matrix $\min \Sigma_{F^\top G} = \min \Sigma_{G^\top F} = \sigma_{G^\top F}^q$, wherein $\sigma_{G^\top F}^q$ represents the $q$-th largest value in the singular value matrix. Ignoring terms representing data, it can be derived that:

$$\mathcal{L}_{ood} - \mathcal{L}_{id} \approx (p_i^2 - p_o^2)(1 - \Sigma_{F^\top G}) + \epsilon > 0, \tag{26}$$

$\square$

**Discussion (Performance difference):** The result intuitively shows that the undirectly learned model performs better on feature distributions with larger sample numbers. As shown in Eq. 7, the difference in model performance between OOD and ID data is linearly related to the proportion of the corresponding samples and the correlation degree between the different feature distributions. What's more, when the out-of-domain data has the same proportion as in-domain data in the training dataset ($p_i = p_o$), or the data distributions of OOD are consistent with ID, the task loss difference between OOD and ID data could be reduced to zero.

## A.3 Proofs for ID-targeted Model Sparse

**Lemma A.6.** *(3.7) Spurious features targeted model sparse can effectively reduce the performance deviation of the learned model between in-domain data and out-domain data.*

$$\left| \frac{\mathcal{R}(X_{ood}) - \mathcal{R}(X_{id})}{\mathcal{R}(X_{ood})^{sparse} - \mathcal{R}(X_{id})^{sparse}} \right| \approx \left| \frac{\sum_{j=1}^m p_o \tilde{\sigma}_j \xi_j \gamma_j X_{ood} - \sum_{i=1}^m p_i \sigma_i \xi_i \lambda_i X_{id}}{\sum_{j=1}^m p_o \tilde{\sigma}_j \xi_j \gamma_j X_{ood} - \sum_{i=1}^\vartheta p_i \sigma_i \xi_i \lambda_i X_{id}} \right| \geq 1, \tag{27}$$

*where $\sigma_i, \tilde{\sigma}_i$ is the i-th maximums in $\Sigma_{E^\top F}$ and $\Sigma_{E^\top G}$. And we have $\sigma > 0$ since the singular values are non-negative. m and $\vartheta$ are the rank of the singular value matrix after performing compact singular decomposition and truncated singular value decomposition on the projections, respectively.*

PROOF OF LEMMA 3.7. As mentioned before, the projection space before the model sparse could be represented as:

$$E\boldsymbol{r} = \sum_{i=1}^m \left( p_{id} \sigma_i \xi_i \lambda_i^\top + p_{ood} \tilde{\sigma}_i \xi_i \gamma_i^\top \right) \tag{28}$$

SFP prunes the model by trimming the smallest singular values in $\Sigma$ as well as their corresponding left and right singular vectors. In this way, SFP could remove the spurious features in ID data space and substructures in the model space simultaneously in a targeted manner

along the directions with weaker actions for projection. The projection space after sparse with only the most important $\vartheta$ singular values can be formalized as:

$$Er^{sparse} = \sum_{i=1}^{\vartheta} p_{id} \sigma_i \xi_i \lambda_i^{\top} + \sum_{j=1}^{m} p_{ood} \tilde{\sigma}_j \xi_j \gamma_j^{\top}. \tag{29}$$

Based on the representation of the projection spaces, the model response to data features $\mathcal{R}(X) = ErX$ can be calculated as:

$$\mathcal{R}(X) = \left\{ p_i \Xi \Sigma_{E^{\top}F} \Lambda^{-1} + p_o \Xi \Sigma_{E^{\top}G} \Gamma^{-1} \right\}^{\top} X^{\top} \tag{30}$$

$\square$

## A.4 Proofs for the correspondence between model substructure and spurious features

Specifically, we define $f^l(x)$ as the feature maps output of $x$ at layer $l$. It represents the projection of $x$ onto the model space defined over the spanning set $E$ to be learned. We abbreviate the final probabilities as $f(x)$ for simplification. Referring to Sec. 3.2.1, we have $x \in X_{id}$ if $\mathcal{L}_{ce}(x) \leq \Delta$. Thus, the optimization target of SFP can be formulated as:

$$\min_E \mathbb{E}_{x \sim X} \mathcal{L}_{ce}(x, \mathcal{W}) + \eta \sum_{l=1}^{L} \mathbb{E}_{x \sim X_{id}} ||f^l(x)||_1, \tag{31}$$

where $\eta$ is the sparsity factor imposed on the feature projections for the identified ID data. It serves as an adjustable weight to calibrate the feature response of ID data, as well as sparse the corresponding substructures.

**Lemma A.7.** *(3.8) Define $e = |f^*(x) - f(x)|$ as the $l_1$-norm between the true distibution $f^*(x)$ and $f(x)$. When $\eta < 2e$, SFP could effectively reduce the learning of the model to spurious features but keep the performance on the same features.*

PROOF OF LEMMA A.7. The prediction errors of feature projections $L_f$ can be defined as:

$$\begin{aligned}
L_{ce} &= |f^*(x) - f(x)|^2 \\
&= \sum_{i,j=j_1 \cup j_2} (f^*(x) - \sigma_{i,j_1} \xi_i^{\top} \lambda_{j_1} - \sigma_{i,j_2} \xi_i^{\top} \gamma_{j_2})^2,
\end{aligned} \tag{32}$$

and the corresponding gradient is:

$$\begin{aligned}
\frac{\partial L_{ce}}{\partial \sigma_{i,j_1} \xi_i} &= \frac{\partial e^2}{\partial \sigma_{i,j} \xi_i} = 2e \frac{\partial e}{\partial \sigma_{i,j} \xi_i} \\
&= 2e \frac{\left| f^*(x) - \sigma_{i,j_1} \xi_i^{\top} \lambda_{j_1} - \sigma_{i,j_2} \xi_i^{\top} \gamma_{j_2} \right|}{\partial \sigma_{i,j} \xi_i} \\
&= -2e \lambda_{j_1},
\end{aligned} \tag{33}$$

where $i$ and $j$ are the index of column vectors in the orthogonal basis for model space and feature space, respectively. For OOD data, the gradient of the column vectors in the OOD projection matrix interacting with the $j_{th}$ feature vector is $-2e\gamma_{j_2}$.

Therefore, for all data samples in the training set, the update of the $i_{th}$ direction vector of the projection matrix at round $t$ is:

$$\sigma_{i,j} \xi_i^t = \sigma_{i,j} \xi_i^{t-1} - p_i(-2e\lambda_{j_1}) - p_o(-2e\gamma_{j_2}) + p_i \eta \lambda_{j_1} \tag{34}$$

Split the in-domain features into the spurious features $F'$ and the invariant features $IN$, and split the out-of-domain features into the unknown features $G'$ and the invariant features $IN$. Since the environment features in-domain and out-domain are different with high probability under the OOD setting, we suppose $F'$ and $G'$ are orthogonal and define $a, b \in \Xi$ as the column vectors interact with $F'$ and $G'$ respectively. The updates of $a, b$ could be formulated as:

$$\begin{aligned}
\sigma_{a,F'} \xi_a^t &= \sigma_{a,F'} \xi_a^{t-1} - p_i(-2e\lambda_{F'}) - p_i \eta \lambda_{F'} \\
\sigma_{b,G'} \xi_b^t &= \sigma_{b,G'} \xi_b^{t-1} - p_o(-2e\gamma_{G'})
\end{aligned} \tag{35}$$

Also, define $c \in \Xi$ to be the set of the column vectors in the projection matrix that interacts with invariant features that are consistent in domain and out of domain, and the updates of $c$ can be computed as:

$$\sigma_{c,IN} \xi_c^t = \sigma_{c,IN} \xi_c^{t-1} + 2ep_i \lambda_{IN} + 2ep_o \gamma_{IN} - p_i \eta \lambda_{IN} \tag{36}$$

To achieve spurious features-targeted unlearning and invariant features-targeted learning of the model, the following constraints need to be satisfied:

$$\begin{aligned}
&2ep_i \lambda_{IN} + 2ep_o \gamma_{IN} - p_i \eta \lambda_{IN} > 2ep_o \gamma_{G'} \\
\Rightarrow\ &\eta \leq \frac{2ep_i \lambda_{IN} + 2ep_o \gamma_{IN} - 2ep_o \gamma_{G'}}{p_i \lambda_{IN}} \approx 2e
\end{aligned} \tag{37}$$

Since the de-learning rate of the spurious feature is positively correlated with $\eta$, the upper bound $\eta = 2e$ is taken in this work. $\square$

## B   ABLATION EXPERIMENTS

In this section, we mainly focus on two aspects: the initialization of the dense model, and the mapping relation versatility: background-label mapping relation in the biased samples' setting.

### B.1   The initialization of the dense model

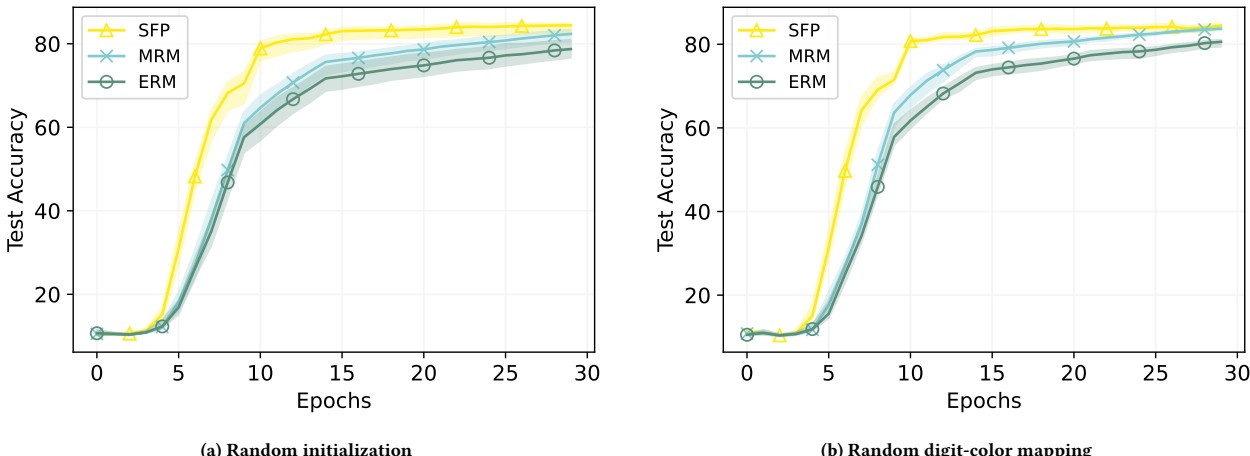

(a) Random initialization                                        (b) Random digit-color mapping

**Figure 8: The effect of some random settings on the performance of SFP.**

First, to demonstrate the consistent performance of the proposed SFP regardless of the randomness of the experiment environments, we used ten different random seeds to initialize the deep learning model and record the final accuracy on FullColoredMNIST (FCMNIST). We use $(0.9, 0.7, 0.0)$ as the biased ratio coefficient in this experiment. Fig. 8a illustrates the mean accuracy of 10 different experiment settings. We notice that the proposed SFP outperforms MRM and ERM in all datasets. This indicates that SFP can successfully sparse the spurious feature-associated network structure regardless of different model initializations.

### B.2   The mappings relations of OOD samples

In our experiment setting, we use the biased data samples, which have a static one-to-one digit-color relationship, as the ID data samples. On the contrary, the OOD data samples have randomly assigned colored backgrounds. To further demonstrate that our method can successfully prune spurious features regardless of the mapping relations, we evaluate the test accuracy in 5 different mapping relation settings and draw the mean accuracy in Fig. 8b. As the experiment result shows, the average accuracy of SFP is relatively higher, and the variance of the accuracy is relatively lower, which shows the superiority of the proposed SFP is stable and robust. This suggests that SFP successfully prunes the sub-network associated with any spurious feature.

### B.3   The feature responses of spurious correlations

Furthermore, to validate the effectiveness of SFP in suppressing the learning of spurious features, we examine the progression of the network's feature responses to in-domain samples across the entire training trajectory. The response values are measured by the average attention across all feature channels at each layer. Specifically, we introduced a channel attention mechanism, named Squeeze-and-Excitation (SE) module [13], to score the channel saliency of feature maps for input $x_i$. The computed channel saliencies, denoted as $\pi_l(x_i)$, are numerical values produced by a Sigmoid function, ranging from 0 to 1. For models trained with ERM on unbiased data, the expected average attention values for the feature channels at each layer are 0.5. These values represent the relative importance of the corresponding feature channel, with smaller values denoting reduced importance. In summary, the sparsity of channel saliency determines the number of effective filters for structures. For inputs, the mean of these channel attentions indicates the models' fitting degree to the current samples. We conducted the experiments on ResNet-18 and ColoredMNIST.

The results are shown in Fig. 9. As the training progresses, SFP gradually weakens the feature responses to spurious correlated data, while under ERM and MOD methods, this response shows no significant changes. The failure of ERM is attributed to its inclination to learn all correlations indiscriminately to enhance predictive accuracy. On the other hand, the failure of the MOD method, as a structured OOD approach, lies in its utilization of existing pruning techniques without specific enhancements for OOD attributes. These pruning methods often lack feature specificity, meaning they do not consider the correspondence between the structure and features. Consequently, they

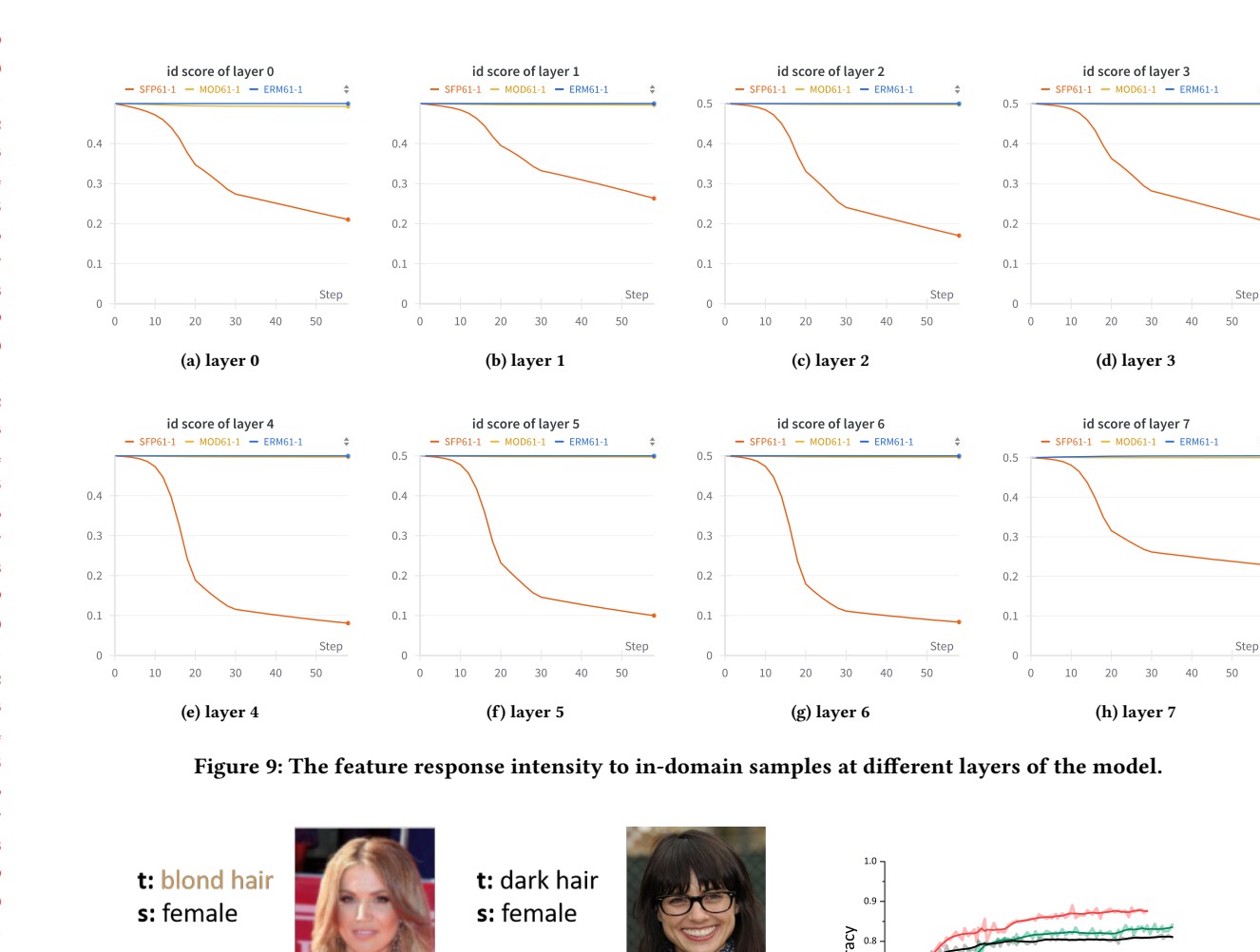

Figure 9: The feature response intensity to in-domain samples at different layers of the model.

(a) Domains

(b) Waterbirds

Figure 10: The average testing accuracy over different domains on CelebA. The hair color blond, dark is used as the target, and the gender male, female is used as the spurious attribute. The smallest combination group is blond-haired males.

apply the same sparse penalty to branches responding to invariant and spurious features simultaneously. In contrast, SFP, designed with OOD attributes in mind, employs feature-specific network pruning. Consequently, it sidesteps the above-mentioned issues.

## C  ADDITIONAL EXPERIMENTAL RESULTS

### C.1  Dataset details

In this section, we will provide a clear description of the non-domainbed datasets in the main paper, including three synthetic dataset - FullColoredMNIST, ColoredObject, and SceneObject, and two real-world datasets - CelebA and Waterbirds.

- **FullColoredMNIST** is a ten-class biased variant of the original MNIST dataset [41]. The digit shapes serve as invariant features while colors as spurious ones. Ten different colors were selected to define a one-to-one corresponding relationship with ten-digit

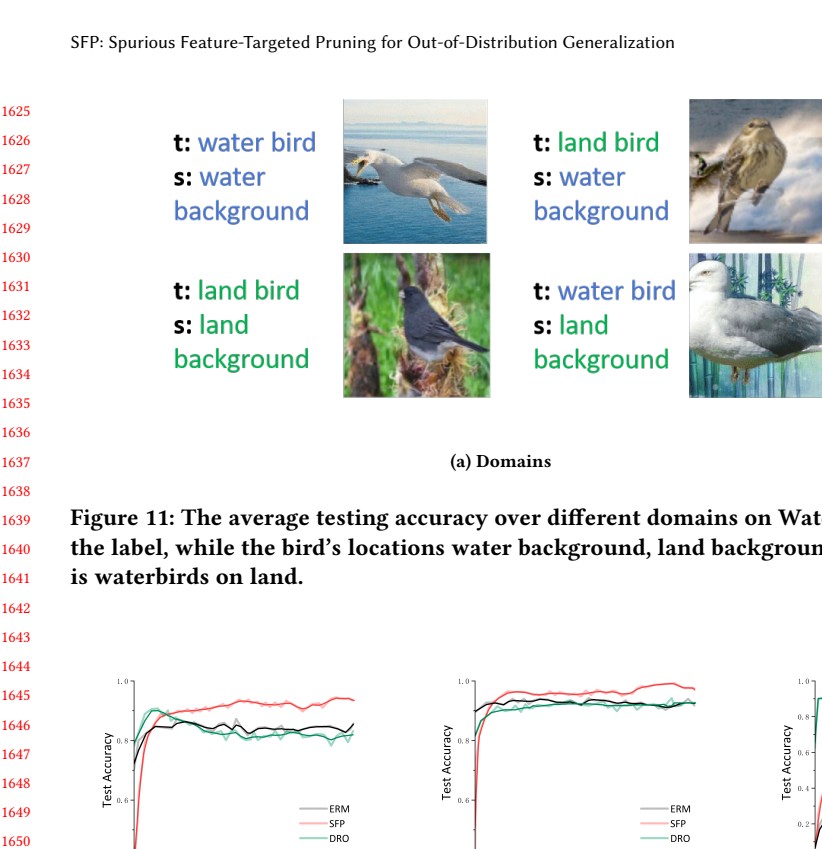
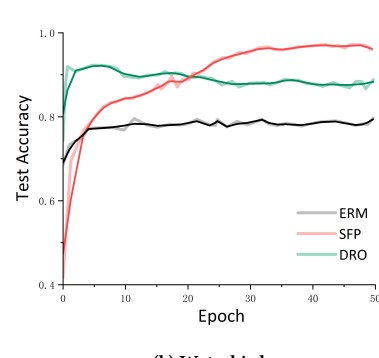

(a) Domains

(b) Waterbirds

Figure 11: The average testing accuracy over different domains on Waterbirds. The bird species waterbird, landbird are used as the label, while the bird's locations water background, land background are used as a spurious attribute. The smallest domain is waterbirds on land.

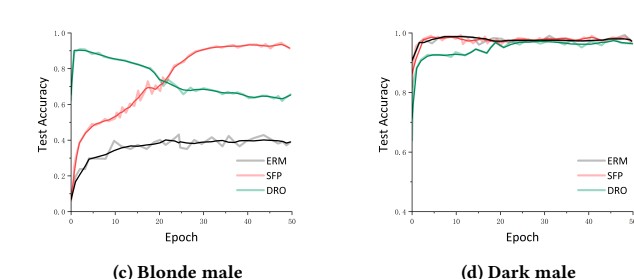

(a) Blonde female

(b) Dark female

(c) Blonde male

(d) Dark male

Figure 12: Testing accuracy of different domains on CelebA.

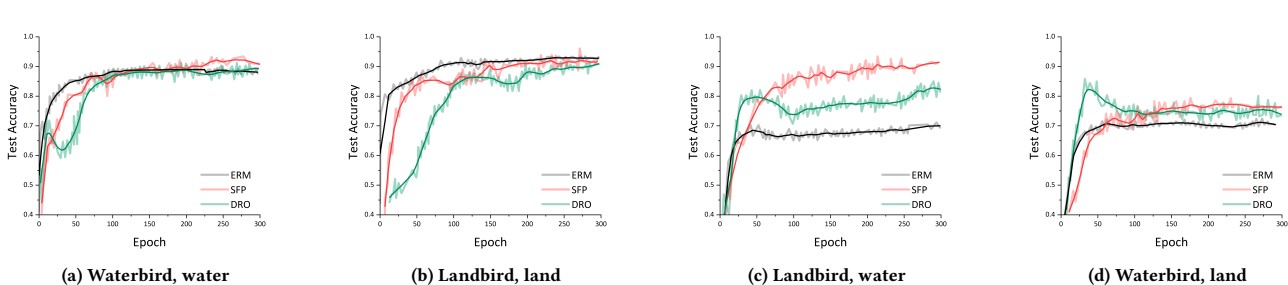

(a) Waterbird, water

(b) Landbird, land

(c) Landbird, water

(d) Waterbird, land

Figure 13: Testing accuracy of different domains on Waterbirds.

classes (e.g., $2 \leftrightarrows green$, $4 \leftrightarrows yellow$). For each domain, a bias coefficient is defined to represent the ratio of images adhering to this specific relationship, with non-conforming images randomly colored.

- **ColoredObject** is constructed by superimposing ten classes of objects extracted from the MSCOCO dataset [22] onto backgrounds of ten distinct colors [41]. These ten classes of objects include boats, airplanes, trucks, dogs, zebras, horses, birds, trains, buses, and motorcycles. The spurious correlation is defined as the one-to-one correspondence between objects and colors.
- **SceneObject** [41] consists of ten classes of objects extracted from the MSCOCO dataset, which are placed into ten scenic backgrounds from the Places dataset [44]. These scenic backgrounds render the task more complex compared to ColoredObject. Similar to FULLCOLOREDMNIST, SceneObject establishes a one-to-one object-scene relationship, making it more biased and consequently more challenging than previous tasks.
- **CelebA** dataset is a widely-used celebrity face dataset with 162770 training examples [25]. It contains 40 attribute labels (like "Smiling", "Wearing Hat", etc.) Following previous OOD works [23, 31, 34], we classify hair color as either blonde or non-blonde, a

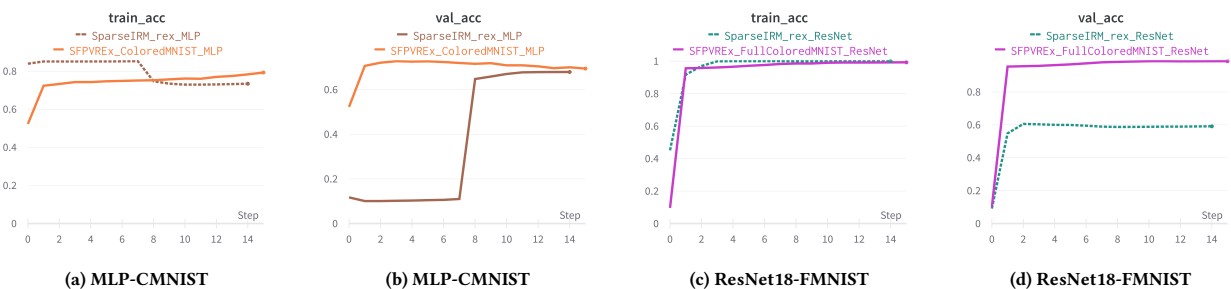

**Figure 14: The accuracy comparison of SFP+IRM and IRM.**

**Figure 15: The accuracy comparison of SFP+VREx and SparseIRM+VREx.**

feature spuriously associated with the gender binary of the celebrities (male or female). The training set is divided into four domains, including drak-haired females, blond-haired females, dark-haired males, and blond-haired males, with 1387 in the smallest group (blond-haired males).

- **WaterBirds** is a subset of the Caltech-UCSD Birds-200-2011 dataset [36] with 4795 training examples, specifically constructed for studying image recognition with spurious correlations of backgrounds [31]. It incorporates images of waterbirds and landbirds from the Caltech-UCSD Birds-200-2011 (CUB) dataset as the foreground, paired with either water or land backgrounds obtained from the Places dataset. The training set is divided into four domains, including landbirds on land, waterbirds on water, landbirds on water, and waterbirds on land, with 56 in the smallest group (waterbirds on land).

## C.2 Evaluation on more datasets

We further expanded the evaluation scope of SFP to two real-world datasets: WaterBirds and CelebA. The experiment is divided into two groups, including comparisons of average accuracy across all domains and comparisons of the accuracy on each individual domain.

Fig. 10 and Fig. 11 illustrated the comparative results of cross-domain average testing accuracy based on the CelebA and Waterbirds datasets, respectively. First, we visualized each domain of CelebA on Fig. 10a, and Waterbirds on Fig. 11a. Subsequently, we compare the cross-domain average accuracies of different methods in Fig. 10b (CelebA) and 11b (Waterbirds). The results demonstrate the superior performance of our proposed method, which reaches a remarkable accuracy of 96.41% on CelebA and 88.13% on Waterbirds. Specifically,

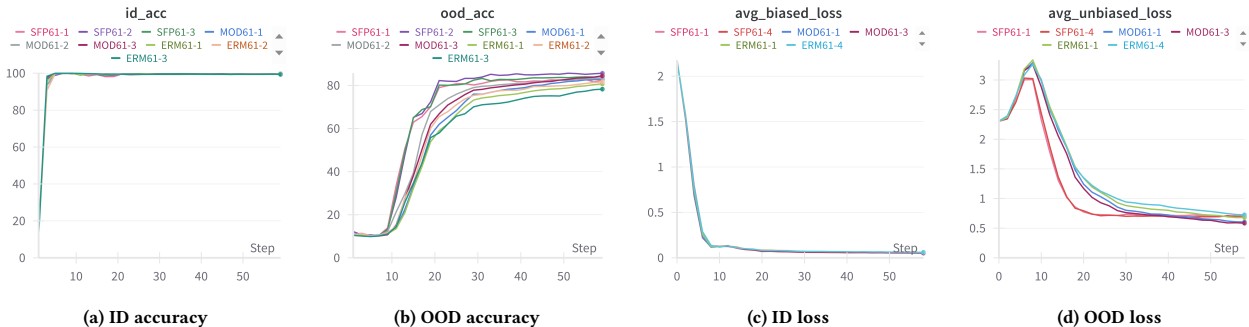

Figure 16: The performance of SFP across various domains with distinct classes.

Figure 17: The evaluation of SFP on different domains.

SFP's cross-domain average accuracy on the CelebA dataset surpasses ERM by 16.8% and DRO by 6.45% (A.b), while on the Waterbirds dataset, it exceeds ERM by 7.23% and DRO by 4.15%.

Further insights into the testing accuracy of individual domains based on the CelebA and Waterbirds datasets are provided in Figures C and D, respectively. Within this framework, the models were trained across all domains while tested on individual domains, with inter-domain sample quantities varied. Specifically, the CelebA dataset encompasses four domains: blonde-haired female, blond-haired male, dark-haired female, and dark-haired male, with the fewest samples found in the blond-haired male domain. Similarly, the Waterbirds dataset consists of four domains: waterbirds on water background, waterbirds on land background, landbirds on water background, and landbirds on land background, with the fewest samples observed in the waterbirds on land background domain. In this context, SFP consistently demonstrates a satisfied performance. As depicted in Fig. 12c and Fig. 12c, SFP achieves a substantial accuracy improvement within the smallest domain, exhibiting a remarkable 42.89% increase on CelebA (with blond male, ERM) and 20.28% increase on Waterbirds (with landbird on water background, ERM). Across the remaining three domains, test accuracies are comparably high. SFP improves the test accuracy by a minor 3.9% over DRO on the blond-haired female domain and 7.02% over ERM on the waterbirds-on-land domain.

**Discussion:** It can be seen that while ERM demonstrates satisfactory performance across multiple domains, it exhibits subpar performance within the smallest domain. What's more, despite DRO achieving an enhancement in average accuracy, it sacrifices performance within specific groups to bolster accuracy within others, exemplifying instances of trade-offs. In contrast, SFP achieves the most robust generalization accuracy by iteratively learning invariant features through feature-oriented model pruning, thereby outperforming the other methods.

## C.3 Evaluation on combined methods

To demonstrate the orthogonal effect of SFP, we evaluate its performance in combination mode with several popular non-structure OOD methods. Firstly, we compare the performance of the most representative IRM and its addition version of SFP. The experiments are conducted on DomainBed, including PACs, COLOREDMNIST, OfficeHome, FullCOLOREDMNIST, and RotatedMNIST. The results are shown in Fig. 14. It can be seen that SFP generally beats the baselines by a large margin and achieves striking results. Specifically, 'SFP + IRM' outperforms IRM with 9.52% on 'PACS-ResNet18', 5.41% on 'COLOREDMNIST-MLP', 0.92% on 'OfficeHome-ResNet18', and 4.45 % on 'RotatedMNIST-MLP'. The experimental results demonstrate that SFP can further improve the performance of non-structure methods in combination mode, confirming its superior orthogonal capabilities.

Additionally, we compare the performance of SFP and the most comparable SparseIRM under combination mode. Fig. 15 presents the performance comparison between SFP+VREx and SparseIRM+VREx. SFP outperforms SparseIRM with 3.41% higher test accuracy on MLP and even 29.12% on ResNet18. An interesting phenomenon is that, on small MLP, SparseIRM first shows overfitting during the training stage, and then, after 7 (× 300 iterations) steps, there is a significant increase in test accuracy. The training process of SparseIRM exhibits an obvious two-stage trend, which is the same as regular non-feature-targeted model pruning. Differently, SFP consistently shows a stable learning curve and achieves higher performance in both ID (train) and OOD (test) environments. In summary, the comparison results demonstrate that SFP has achieved preeminent performance across most structure-based OOD generalization methods.

## C.4 Evaluation on different domains

We further evaluated the performance of the SFP on each domain to explore whether varying invariant targets under the same intensity of spurious features affect OOD generalization. The results are presented in the Fig. 16. We first illustrate several examples from the training and testing sets. In the training set, most samples align with the previously described in-domain data settings, establishing a one-to-one relationship between the background and the label for each digit. Each testing set contains digits of only one category, with the background of that digit category differing from the training domain.

In this setup, we evaluated the performance of SFP on different test domains and compared the results with the ERM method and another popular structured method MRM. Firstly, vertically within the same graph, SFP achieved better generalization performance than MRM in almost all cases. Secondly, horizontally across different graphs, there was a significant difference in the accuracy improvement of SFP over ERM across different classes. In examples where invariant features such as digits 0 and 1 are relatively easy to learn, the three methods showed comparable accuracy. This indicates that ERM and MOD also learned invariant features well. However, in examples where invariant features such as digits 5 and 8 are more challenging to learn, SFP outperformed MOD and ERM significantly. This not only highlights the strong OOD generalization performance of SFP but also reflects the poor mastery of ERM in complex invariant feature scenarios. The experimental conclusions in this section align with previous works [4, 26]: neural networks trained with the ERM inherently learn both invariant and spurious features, but they tend to prioritize shortcuts at the early stage.

Similarly, we also split the dataset as in-domain and out-of-domain and tracked the performance separately during training. The results are shown in Fig. 17. We conducted two evaluations with different initializations for each method. It can be observed that in all cases, SFP achieved outstanding OOD generalization performance, surpassing the baseline methods by approximately 5% during convergence. Moreover, we compared the task loss of SFP and other methods across different data domains, and the results further validated the superior performance of the proposed approach.

