# OpenReview forum: "SFP: Spurious Feature-Targeted Pruning for Out-of-Distribution Generalization"
_acmmm.org/ACMMM/2024/Conference — MM2024 Poster_

### Official Review · Reviewer_f3oV · 2024-04-28

**Rating:** 5
**Confidence:** 3

**Summary:**

This work proposes a new method of Spurious Feature-targeted Pruning (SFP) for out-of-distribution (OOD) generalization by inducing invariant substructures across sparse samples. The authors introduce a threshold Δ acts as the identification criterion for spurious features. Based on this, they weaken the spurious feature projections onto the model space to prevent overfitting. This paper also provides a theoretical understanding that the proposed method reduces the performance deviation between in- and out-of-domain samples. Experiments on multiple benchmark datasets demonstrate the validity of SFP for OOD generalization.

**Strengths:**

1. The paper is interesting; it has clear motivation and is easy to follow

2. This work provides a theoretical analysis of the SFP and OOD generalization

3. Extensive experiments on multiple datasets, including the combined methods and different domains

**Limitations:**

1. The author performs SVD of the feature projections into the model space. If the dataset and feature dimension increase, the time complexity will also increase exponentially. It would be good to provide an explanation for this problem
#

2. Although it was recently published, it would be good to add a comparison with some papers [1], [2]. Especially, [1] applied pruning for domain generalization based on L2. Does the SVD-based algorithm have a tighter bound than this method?
* [1] Pruning for Better Domain Generalizability, ICML '23
* [2] SPURIOUS FEATURE DIVERSIFICATION IMPROVES OUT-OF-DISTRIBUTION GENERALIZATION, ICLR '24

3. Some typos, e.g., FTG in Eq. 8~11 and f(x) in Eq. 17

**Suitability:**

2

---

### Official Review · Reviewer_jmVJ · 2024-05-26

**Rating:** 4
**Confidence:** 2

**Summary:**

This paper proposes a Spurious Featuretargeted Pruning framework to induce the authentic invariant substructures without referring to the above concerns. Experiments on various OOD datasets show that SFP can significantly outperform both structure-based and non-structure-based OOD generalization state-of-the-art (SOTA) methods by large margins.

**Strengths:**

1. this paper is well-written and easy to understand
2. solid theoretical results
3. rich visualization

**Limitations:**

1. This paper lacks comparison with recent work [1,2,3] empirically and methodologically.


[1] Moderately Distributional Exploration for Domain Generalization
[2] Causality-inspired representation learning for domain generalization
[3] A fourier-based framework for domain generalization

**Suitability:**

2

---

### Official Review · Reviewer_sX8t · 2024-05-29

**Rating:** 4
**Confidence:** 3

**Summary:**

This paper proposes SFP, a novel spurious feature-targeted pruning framework. Specifically, SFP distinguishes spurious features with a threshold and penalizes the corresponding feature. This paper is novel and well-organized. My concerns mainly focus on the experiment results.

**Strengths:**

1.	The theoretical analysis in this paper is well-organized.
2.	The experiments in this paper are comprehensive. This paper conducts experiments on various datasets and compares SFP with the latest baselines. Moreover, the case studies in this paper are interesting.

**Limitations:**

First, as the results in the column “Average” show, the basic ERM outperforms all the OOD baselines. This phenomenon is very strange, making me doubt the correctness of the code implementation in this paper.
Second, the worst baseline MLDG is underlined. What does the underlined result here mean?
Third, what do the results in grey mean here? Does it mean these results are worse than ERM, which is consistent with the setting in Fishr [1]? If so, why are so many methods inferior to ERM while not marked in grey?
These results are quite strange and confusing, further making me doubt the proposed method's effectiveness. If the author can not answer the above questions, I tend to decrease my scores for this paper.


[1] Rame, Alexandre, Corentin Dancette, and Matthieu Cord. "Fishr: Invariant gradient variances for out-of-distribution generalization." International Conference on Machine Learning. PMLR, 2022.

**Suitability:**

2

---

### Meta-Review · Area_Chair_bA55 · 2024-06-28

**Recommendation:** Accept (Poster)
**Confidence:** 4

**Metareview:**

This paper induces the authentic invariant substructures without referring to the above concerns when solving the OOD generalization problem. Experiments also show that the method can outperform other SOTA methods by large margins.
The authors also give positive feedbacks after the rebuttal phase, giving 2 borderline acceptance and 1 weak acceptance. To this end, I recommend accepting this paper.